# Trace element concentrations in forage seagrass species of *Chelonia mydas* along the Great Barrier Reef

Adam Wilkinson[1]*, Ellen Ariel[1], Jason van de Merwe[2], Jon Brodie[3]

**1** College of Public Health, Medical and Veterinary Sciences, James Cook University, Townsville, Queensland, Australia, **2** Australian Rivers Institute and School of Environment and Science, Griffith University, Gold Coast, Queensland, Australia, **3** ARC Centre of Excellence for Coral Reef Studies, James Cook University, Townsville, Queensland, Australia

\* adam.wilkinson@my.jcu.edu.au

**Data Availability Statement:** All relevant data are within the paper and its Supporting information files.

**Funding:** The author(s) received no specific funding for this work.

## Abstract

Toxic metal exposure is a threat to green sea turtles (*Chelonia mydas*) inhabiting and foraging in coastal seagrass meadows and are of particular concern in local bays of the Great Barrier Reef (GBR), as numerous sources of metal contaminants are located within the region. Seagrass species tend to bioaccumulate metals at concentrations greater than that detected in the surrounding environment. Little is known regarding ecotoxicological impacts of environmental metal loads on seagrass or *Chelonia mydas* (*C. mydas*), and thus this study aimed to investigate and describe seagrass metal loads in three central GBR coastal sites and one offshore site located in the northern GBR. Primary seagrass forage of *C. mydas* was identified, and samples collected from foraging sites before and after the 2018/ 2019 wet season, and multivariate differences in metal profiles investigated between sites and sampling events. Most metals investigated were higher at one or more coastal sites, relative to data obtained from the offshore site, and cadmium (Cd), cobalt (Co), iron (Fe) and manganese (Mn) were found to be higher at all coastal sites. Principle Component Analysis (PCA) found that metal profiles in the coastal sites were similar, but all were distinctly different from that of the offshore data. Coastal foraging sites are influenced by land-based contaminants that can enter the coastal zone via river discharge during periods of heavy rainfall, and impact sites closest to sources. Bioavailability of metal elements are determined by complex interactions and processes that are largely unknown, but association between elevated metal loads and turtle disease warrants further investigation to better understand the impact of environmental contaminants on ecologically important seagrass and associated macrograzers.

## Introduction

Sea turtles are air-breathing reptiles. Like marine mammals, sea turtles have lungs and must return to the surface to breathe [1]. Due to this, the primary source of metal element exposure

**Competing interests:** The authors have declared that no competing interests exist.

for green turtles (*Chelonia mydas*) is through ingestion of their diet [2]. *Chelonia mydas* (*C. mydas*) has a complex life history that includes a diverse diet, dependent on life phase [3]. Juveniles migrate to the coastal zone, inhabiting foraging grounds, where individuals undergo an ontogenetic dietary shift from a carnivorous to an herbivorous diet [3]. Once herbivorous, *C. mydas* forage on a range of material, dependent on region and what forage species are most predominant [4]. Whilst macroalgal species are common as primary forage material in *C. mydas* diets in some regions [4], at coastal sites along the Great Barrier Reef (GBR), *C. mydas* feed primarily on seagrass species [5].

In addition to providing forage for *C. mydas*, seagrass fulfil several integral ecological functions, such as sediment stabilisation [6–9], nutrient cycling [10, 11], the sequestration of carbon [12–14], as nursery grounds and foraging material for a wide range of marine organisms [6, 15]. Most seagrass species grow in the coastal zone, often near anthropogenic activity and potential marine contamination sources [16, 17]. Declines in seagrass distribution has been widespread in recent decades [18–20].

Several seagrass species are considered reliable bioindicators for the health of an ecosystem, as an early warning of any elevated contaminants, or decline in water quality parameters [21, 22]. Seagrass are efficient at accumulating metal concentrations, at magnitudes higher than that detected in the surrounding water [23]. Metals are highly persistent and remain in the environment indefinitely, binding to fine particulate sediment and settling on the substrate where concentrations are absorbed by seagrass [5]. Potential bioaccumulation of metals in seagrass may be a significant exposure pathway for turtles to elements at toxic concentrations [5]. *C. mydas* display strict site fidelity within a small local foraging range (2–3 km$^2$), and likely inhabit the same seagrass meadow for long periods, regardless of environmental condition and water quality, even if environmental health deteriorates [24].

Metal contamination of marine ecosystems has increased significantly in recent years, with new chemicals frequently being introduced through industrial and agricultural processes and sequestered metal loads remobilised and redistributed due to dredging events and environmental disturbances such as flooding and cyclone activity [5, 25]. Some metals occur naturally in the marine environment, many of which are deemed essential elements for numerous biochemical and physiological processes (such as iron, Fe; copper, Cu and magnesium, Mg) [26]. However, essential elements may have toxic effects if concentrations exceeding optimal thresholds are experienced (as well as extremely low levels), particularly over long periods of time. For example, Fe, Cu and zinc (Zn) may cause reduced immune function in marine organisms when elevated [26]. Conversely, numerous metal elements are non-essential for life and are often toxic to organisms at very low concentrations [27, 28]. Non-essential elements commonly detected in the marine environment include, cadmium (Cd) and lead (Pb), with elements such as cobalt (Co) being understudied, but potentially toxic to marine turtles [5]. Metals of most concern are those elements (essential and non-essential) that cause known toxic effects to immune function and biochemical processes, such as Cd and Pb [26, 29].

Metal contamination (essential and non-essential) of coastal zones occurs via several exposure pathways. Freshwater runoff of sediments, during periods of heavy rainfall and atmospheric deposition of metal particles, are the main transportation pathways of metals to the marine environment [30–32]. Furthermore, anthropogenic pressures and processes such as mining, metal refining, agricultural chemical application, dredging and drainage of industrial waste, are able to change the distribution and composition of any geoavailable and bioavailable metal concentrations within the coastal zone [33].

This study aimed to describe seagrass (determined by identification of local *C. mydas* primary foraged species) trace metal concentrations at several coastal sites along the Great Barrier

Reef (GBR), which are important foraging grounds for *C. mydas* and other macro grazers (such as dugongs) [34]. Such investigation is significant as it provides data on the prevalence of ecologically relevant metals (deemed toxic or are commonly measured in studies with similar scope) in the region. Metals of focus in this study included non-essential elements, cadmium (Cd), cobalt (Co), nickel (Ni) and lead (Pb), and several essential elements (e.g. Fe, Cu and Zn), capable of affecting immune function of marine turtles at ecologically relevant concentrations [26].

## Materials and methods

In this study primary foraged seagrass species of local *C. mydas* populations was first determined by conducting gastric lavage sampling on a subsection of each population (up to 20 individuals from each study site). Rather than directly analysing the gastric lavage samples for metals, a total of 82 seagrass samples were collected by hand, either during dedicated field work, or during other sampling efforts (turtle capture and sampling). The reasoning for collecting fresh seagrass samples as opposed to analysing the gastric samples, was to ensure minimum contamination, and to allow for a wider area of each foraging ground to be sampled. Additionally, sampling known forage species meant that analysis could be completed prior to, and following, the wet season, without having to recapture individual green turtles. Once identified, seagrass samples were analysed for a suite of 10 metal elements using inductively coupled Plasma Optical Emission Spectrometry (ICP-OES) to describe concentrations detected within coastal meadows along the Central GBR. Multivariate analysis informed investigation into trace metal profiles in species of local bays before and after the 2018/19 wet season.

### Study sites

Three geographically distinct study sites were sampled along the east coast of Australia, adjacent to the GBR (Fig 1). Firstly, Cockle Bay (CB) (19 º 10 ' 26.7 "S 146 º 49 ' 32.1"E) is a westerly facing bay of Magnetic Island, 8 km off the coast of Townsville, Queensland and forms a part of Cleveland Bay. Industry practices such as metal processing (including Zn, Cu and Ni), urban runoff from the city of Townsville (population about 200,000), and major sea port practices (including regular channel dredging) take place within the area [35]. Secondly, Upstart Bay (UB) (19˚44 ' 44.4"S 147˚36 ' 03.8 "E) is a north facing bay, receiving river discharges from the major catchment of the Burdekin River (which also influences CB to a lesser extent), dominated by agricultural and grazing practices, and with a prominent mining background, located 150 km south of Townsville. The Burdekin catchment is one of the two largest GBR catchments (the other being the Fitzroy) with an area of 140,000 km$^2$. Finally, Edgecumbe Bay (EB) (20˚ 6' 49'' S, 148˚ 23' 25'' E) is located south of Bowen, Queensland, approximately 200 km south of Townsville. Within the catchment draining into EB there are a number of point and non-point sources of potential contaminants, including a wastewater treatment plant (for the town of Bowen–population of approximately 10,000), cokeworks and sugarcane farms (mostly on the catchment of the Gregory River in the south of the bay), and rarely, from discharge plumes from the Burdekin River [36]. Seagrass metal data [5] from a fourth site, the Howick Island Group (HWK), a mid-shelf group of remote reefs found in the northern region of the GBR (14 º 30 ' 11"S 144 º 58 ' 26 "E), was also included. HWK is likely to be minimally influenced by anthropogenic activity, with limited exposure to land-sourced contamination due to geographical proximity. The study location is located over 130 km from the nearest human settlement (Cooktown) and at least 20 km offshore from the coastal zone of the mainland.

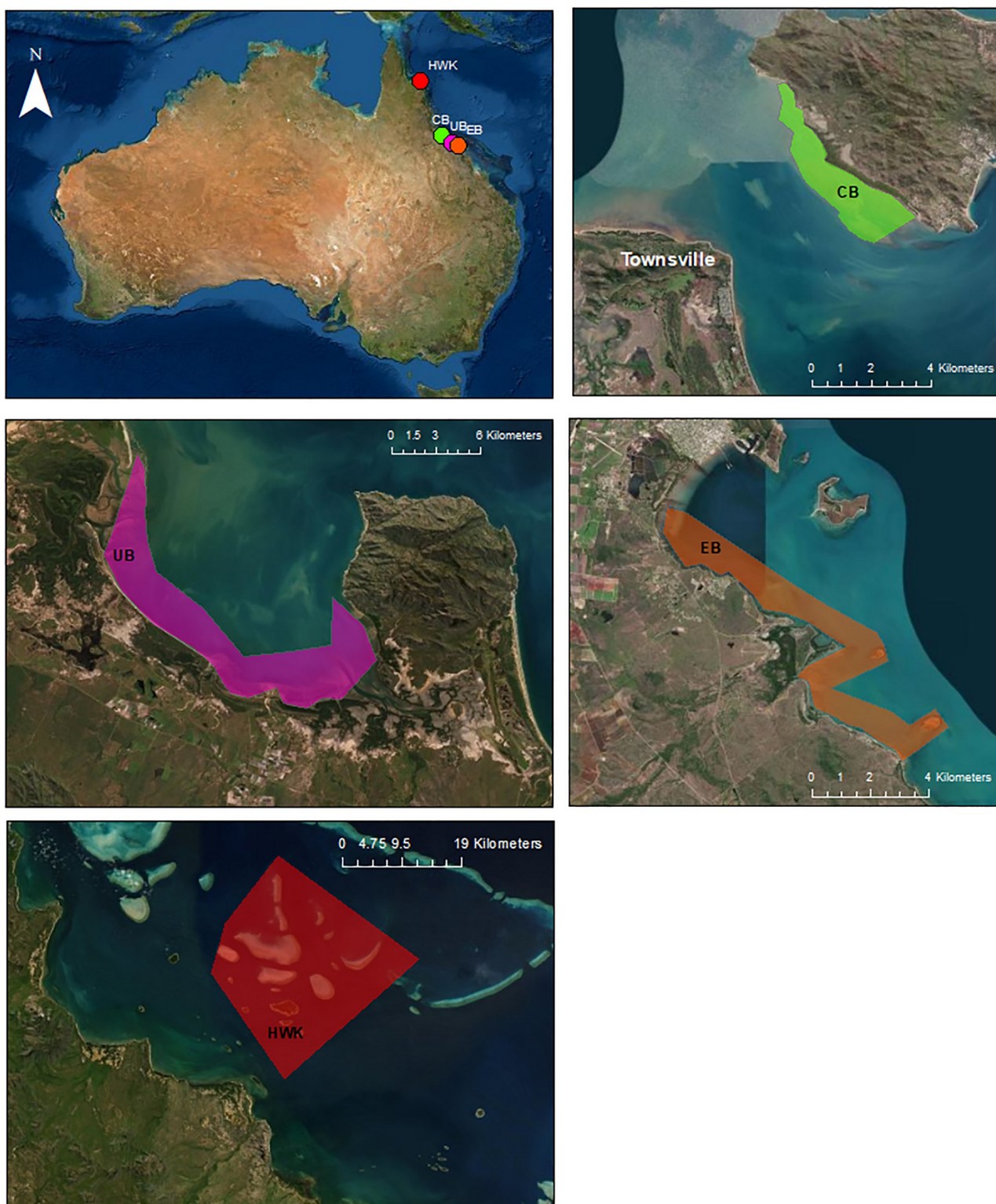

**Fig 1. Overview of all study sites sampled.** Map of three study areas (Cockle Bay (CB), Upstart Bay (UB) and Edgecumbe Bay (EB)) where preferred green turtle seagrass forage was sampled, and the offshore site from which data was also analysed (Howick Island Group, HWK). Each site is colour coordinated between the national overview and localised maps. Areas highlighted in colour define local green turtle foraging ranges. The areas highlighted with white and black depict seagrass sampling locations at each study site. The World imagery applied as the base map in this figure is attributed to Esri and modifications were conducted using ArcMap (version 10.7.1, Esri. ArcGIS and ArcMap, California, USA). and is permitted for use as per Esri terms of service and the DMCA (Digital Millennium Copyright Act).

## Turtle capture and sampling

In total, 46 turtles were captured using turtle rodeo techniques described in [37]. Individually numbered titanium flipper tags (Department of Environmental Sciences, Queensland Government) were applied, as described by Eckert et al. [38]. Curved carapace length (CCL), from the notch of the supracaudal scute to the line where skin joins the anterior edge of the carapace, along the midline ridge of the carapace, was measured using a flexible tape measure (cm), to the accuracy of ± 0.1 cm. Large barnacles were removed from the carapace with long nose plyers if their position obstructed accurate CCL measurements.

Research was carried out under all necessary permits from James Cook University Animals Ethics Committee (A2396), Department of Environment and Science (WISP18586417 and WISP18596817) and Great Barrier Reef Marine Parks Authority (G17/39429.1).

## Gastric lavage

A total of 46 *C. mydas* stomach contents were collected, across all sites (CB = 14, UB = 20 and EB = 12), by modifying the protocol outlined in [39]. Briefly, a custom-made water pumping mechanism was designed, assembled, and tested by experienced personnel prior to use in the field. A foot pump (Whale babyfoot pump, Whale Marine, Northern Ireland) was connected to a water intake hose and an outtake hose, made from polyurethane tubing (8 mm diameter, with the ends melted and rounded to prevent injury to the turtle's digestive tract). Firstly, captured turtles were elevated, with their head at the lowest point and secured in a fixed position by trained personnel. Turtles were encouraged to open their mouth by applying gentle pressure between the jaws, and once open, a wooden bit was inserted across the mouth to prevent closing. A 15–20 cm length of polyvinyl chloride tubing (10 mm diameter) was inserted into the mouth to offer stability for the insertion of the water tube (150 cm long and 8 mm diameter, marked at every 10 cm for the first 100 cm), which was lubricated with olive oil and slowly inserted down the digestive tract. The tube was slowly rotated during insertion to aid in the breaking up of any food bolus (obstruction of pre-digested material) in the throat. Once the marking, indicating insertion to 50–70 cm was achieved (determined on a case-by-case basis, dependant on turtle size), a constant flow of untreated sea water (1 L per minute) was initiated by regular use of the foot pump. Sea water was brought into the system by the intake tube, from a clean bucket filled with local sea water. As the water drained from the turtle all forage material was collected using a sieve (310 μm mesh size) positioned below the mouth. The procedure was conducted for no longer than five minutes and the turtle was maintained in the head down position until all water was drained. All equipment was sterilised using hospital grade detergent (benzalkonium chloride), and thoroughly rinsed between turtles. Samples were then placed on ice until return to the lab, where they were stored at -20 ˚C until identification and analysis.

## Forage sample species identification

Forage material was thawed and separated out into individual species. To identify the seagrass species that were being consumed by *C. mydas* at the study sites, forage species were visually identified by experienced personnel using taxonomical identification guides, where necessary. Forty out of the total 46 (87%) gastric lavage samples collected contained the seagrass species, *Halodule uninervis* (*Halodule*), and the remaining six samples (from CB only, equivalent to 42.3% of CB samples) predominantly contained *Cymodocea serrulata* (*Cymodocea*). Either one of these seagrass species commonly made up 100% of the biomass of an individual lavage sample. Occasionally, in samples collected from CB included small proportions (< 10% of overall biomass) of red algae, however taxonomic identification to the species level was not conducted

here as low abundance did not warrant further study. The identification of forage species informed the collection of seagrass species for the metal concentration analysis.

## Seagrass collection

Seagrass sampling at all coastal sites was conducted before (July–October 2018) and after (February–June 2019) the 2018/19 wet season. The total number of samples collected per study site were as follows: n = 16 pre-wet and n = 12 post-wet season in CB, n = 15 pre-wet and n = 15 post-wet season in UB and n = 11 pre-wet and n = 13 post-wet season in EB. HWK Seagrass sampling was undertaken in July and August of 2015 and 2016. As *Cymodocea* was only present in lavage samples from CB green turtles, this seagrass species was only collected from CB. Samples were collected from the intertidal and subtidal zones within known turtle foraging grounds (identified through turtle sampling events). Personnel wore nitrile gloves when handling samples to minimise cross-contamination. Samples were collected at least 100 meters (max 300 m) apart, parallel to the shoreline, either on foot during low tide, or using snorkel techniques (max depth of approximately 2.5 m). Approximately 60 grams of above- and below- ground material (leaves and rhizomes) were collected and placed in food grade zip lock bags and stored on ice until return to the laboratory whereby samples were stored at -20 ˚C, until processing and analysis. Eighteen samples from HWK made up the offshore data provided by Thomas et al [5] and were collected in a similar way, by hand at low tide.

## Seagrass sample preparation

Prior to analysis, all seagrass samples were thawed, and above and below ground material was separated, and leaves were removed. Small epiphytes growing on the leaf surface were included in the sample, as turtles foraging on seagrass would ingest such epibionts along with the intended forage species. All large debris (shells, shale, sand, etc.) was rinsed off each sample prior to drying, using fresh water. Wet weights were recorded for each sample prior to being oven dried for 48 hr at 60 ˚C. Dry weights of each sample were then recorded before homogenising into a fine powder using a pestle and mortar (sterilised with ethanol in between samples). A minimum of 200 mg of homogenised material (per sample) was submitted to the Advanced Analytical Centre (AAC, James Cook University) for acid digestion and ICP-OES analysis.

## ICP-OES analysis

A suite of 10 metals (aluminium, Al, Cd, Co, Cu, Fe, Mg, Mn, Ni, Pb and Zn) were analysed in each sample. Seagrass samples were digested using a microwave assisted digestion oven (Bergof SW-4). A total of 100 mg of each sample was placed into the digestion vessel. Next, 4 mL SupraPure (Merk Germany) double distilled $HNO_3$ and 1 mL AR Grade $H_2O_2$ were added into the vessel. The sample solution was kept in the fume hood for 2 h until the reaction was complete. Vessels were loaded into the microwave oven and heated to 185˚C for 10 minutes. Once cooled, 150 mL of the digested samples were transferred to volumetric flask and diluted 50-fold, with Milli-Q water. Inductively Coupled Plasma Optical Emission Spectrometry (ICP-OES) was conducted using the Agilent 5100-ICP (Agilent Technologies, USA). External calibration strategy was used by applying a series of multi-element standard solutions containing all the elements of interest. $HNO_3$ and $H_2O_2$ (reagents used in sample digestion, minus the sample) were included as procedure blanks for all elements and used to calculate the limit of detection values (LOD), which was defined as three times the standard deviation of each element's blanks. Three samples were randomly selected and duplicated to check for consistency. To assure instrument calibration quality, independent standards (1 ppm) were included, with

reported recoveries ranging from 87% (Al) and 103% (Mg). Two Certified Reference Materials (CRMs; GBW07605 Tea Leaves and NIST 1566 Oyster Tissues) were analysed to validate the analytical method and % recoveries ranged from 92% (Cu) to 118% (Cd). All metal concentrations are reported here as mg/kg of sample dry weight (dw).

This study and the reference study analysed seagrass samples using similar but distinct analytical techniques. Thomas et al [5] applied inductively coupled plasma mas-spectrometry (ICP-MS) rather than ICP-OES though both are suitable options for the detection of environmental metal loads in organic material such as seagrass. In ICP-OES, digested samples are nebulised and converted to plasma whereby electrons become excited. During the de-excitation phase, light is emitted from atoms and ions at different wavelengths dependant on metal. Wavelengths are then separated, detected, and quantified. In ICP-MS, ions present in the plasma are divided using mass spectrometry which separates and categorises each element by mass/charge ratio. ICP-MS can detect ultra-trace concentrations but both techniques are ideal for measuring the same suit of metals at ppm (mg/kg). Furthermore ICP-MS has the ability to differentiate between isotopes of the same element, but this was not applicable in the current study as focus was placed on total metal concentrations only.

### Data analysis

Metal concentrations for all samples collected in this study (pre-wet and post-wet season), and reference value data obtained from Thomas et al [5], were reported as mg/kg, and any concentrations found to be below limits of detection (LOD) were considered half of the LOD [40], this same approach was applied in the reference study for HWK data. Pb was removed from further analysis due to having >40% samples <LOD (see S1 Fig). In addition, Mg and Zn concentrations were removed from the data set which included both current data and data collected from HWK, as these elements were not reported by Thomas et al. [5].

Spatio-temporal variation between study locations and sampling events (pre-wet- and post-wet season) was conducted for all sites sampled within this study. Additionally, Principal Component Analysis (PCA) was conducted to measure variation between metal profiles from each coastal site and HWK. The HWK data was collected prior to the wet season (in July/August of 2015–16), and thus for PCA analysis, only the pre-wet season data for the coastal sites was used. To determine the most important dimensions in the data, two dimension-reduction protocols, scree plots and quality of representation measurements (cos2), were employed. Statistical analysis and plotting of PCA was conducted in the R statistical program (R Core Team, 2019), using the R packages 'Tidyverse'(data exploration [41]), 'FactoShiny'(multivariate analysis and plotting) [41] and 'FactoMineR' (Factor analysis) [42].

## Results

### Spatial patterns in metal profiles of preferred green turtle forage

When describing the mean metal concentrations from each coastal site relative to that of HWK, concentrations were generally higher in the coastal sites, for both non-essential elements (Cd, Co, and Ni) and essential metals (Cu, Fe, and Mn), though differences were observed across all coastal sites for Cd, Co, Fe, and Mn only (Table 1).

To investigate whether metal profiles differed spatially, between the three coastal foraging sites and the offshore natural baseline site (HWK), PCA was conducted on pre-wet season data for each site. The scree plot indicated that the first two data dimensions adequately represented most of the variation in the data. Therefore, the suite of eight metals (variables or dimensions) were reduced to two principal components (Dim 1 and 2), which together represented 67.02% of the total variation of the data (Fig 2).

**Table 1. Table of mean seagrass metal concentrations.** Mean concentration (mg/kg dw) and standard deviation (SD) of metal elements at each coastal site (CB, UB and EB) pre-wet season, and data from a foraging ground at the offshore site HWK, provided by Thomas et al [5].

| Element | CB | | UB | | EB | | HWK | |
|---|---|---|---|---|---|---|---|---|
| | Mean | SD | Mean | SD | Mean | SD | Mean | SD |
| Al | 3289 | 2629 | 1726 | 887 | 1846 | 753 | 3020 | 1600 |
| Cd | 0.37 | 0.12 | 0.27 | 0.10 | 0.33 | 0.11 | 0.20 | 0.07 |
| Co | 1.75 | 0.51 | 1.87 | 0.45 | 1.42 | 0.66 | 0.52 | 0.21 |
| Cu | 5.64 | 2.23 | 5.05 | 0.50 | 2.96 | 0.62 | 2.47 | 3.30 |
| Fe | 3382 | 1533 | 1954 | 1147 | 3123 | 1088 | 1696 | 827 |
| Mn | 355 | 73 | 246 | 114 | 208 | 111 | 35 | 7.67 |
| Ni | 3.66 | 1.27 | 4.25 | 1.53 | 3.87 | 1.68 | 3.04 | 0.90 |

Squared cosine ($\cos^2$) indicates the importance of a metal element to a particular principal component (or the quality of representation) and supported the reduction to two principal components. A $\cos^2 > 0.65$ true for all metals in the analysis suite, excluding Al and Ni, indicated that most of the data variation was indeed accounted for by the first two dimensions. Rather than removing Al and Ni from the analysis because of low representation, the elements were maintained for consistency. Further confirmation was observed in the scree plot whereby the elbow was at dimension 3 with only variation accounted for by dimension 1 and 2 being present below the line. The Al cluster (top right) aligns closely to Dimension 2 and the others (Cd-Fe-Ni and Co-Cu-Mn clusters), top right and bottom right) align closer to Dimension 1, and thus influence the respective components, and therefore the entire data set, when analysing the results in a reduced space.

When individual seagrass samples were plotted (rather than variable loadings), all coastal site data were relatively evenly spread across the plot, and no separation between these locations was observed (Fig 3). EB variation was greatest of all sites as demonstrated by largest confidence ellipse and data point spread. All coastal seagrass data was distinctly separate from the HWK data (green), with individual sample loadings being clustered tightly and minimal overlap with coastal site data occurring. Confidence ellipses for each of the three coastal sites overlapped, indicative of similarities in each site metal profiles. CB (black) overlapped with both UB (blue) and EB (red), suggesting more similarity between CB and the other sites and little to none between UB and EB. HWK ellipse was distant from all coast sites while coastal data was more closely congregated, suggesting associations between offshore and coastal metal profiles were less likely than between coastal sites.

## Temporal analysis of seagrass metal profiles between sampling events

Differences in mean seagrass metal concentrations were observed at each coastal site between sampling events (pre-wet and post-wet season) and reported in Table 2. At CB, some element concentrations decreased over the wet season while others increased. Co and Cu were lower post-wet season at CB and EB but higher at UB. Mean Al concentrations increased at CB and EB but declined in UB when comparing between data from pre-wet and post-wet season. Similarly, Fe increased at CB and UB but decreased at EB from before to post-wet the wet season. Conversely, Mn decreased at CB and EB and increased between pre-wet and post-wet season data at UB. Cd and Ni concentrations remained similar at all sites when comparing pre-wet and post-wet season data.

A PCA was conducted to investigate any differences in total metal profile between seagrass samples collected from coastal sites pre-wet and post-wet season, and to identify the metals that most influenced the differences between sampling events (Fig 4). The scree plot

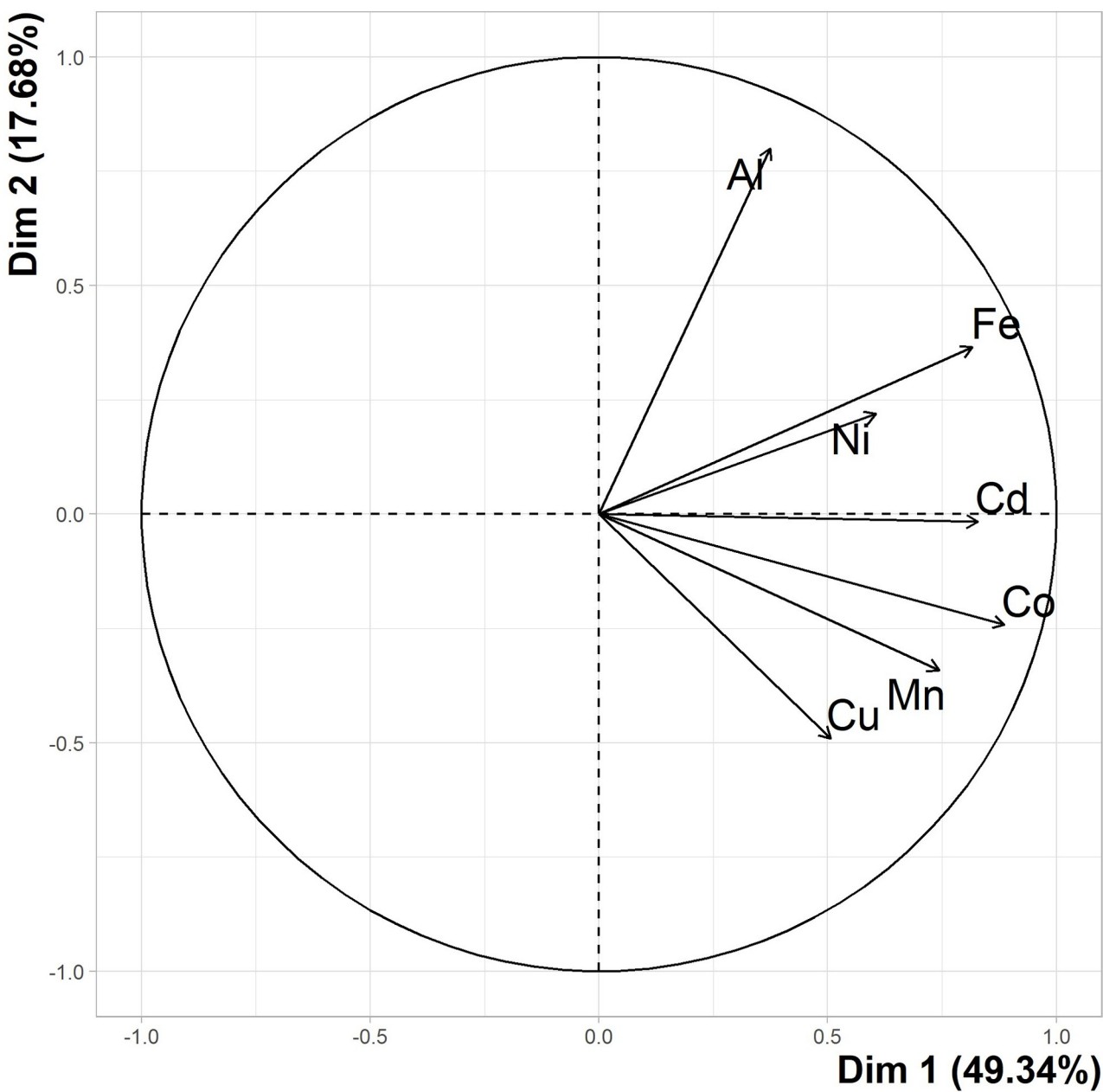

**Fig 2. PCA Variable loading plot.** Plot of the findings of all metal concentration data from seagrass collected from coastal sites of this study and baseline data obtained from HWK and provided by Thomas et al [5]. Dimension 1 (Dim 1) represents 49.34% of total variations and Dim 2 represents 17.68%.

accompanying the analysis determined that much of the data was adequately represented by the first two dimensions (principal components), which represented 57.93% of the variation. Though unlike Fig 2, the first two dimensions did not account for >65% Cos2 of the dataset. However, most elements (Cd, Co, Cu, Fe, Ni and Zn) were represented (>65% Cos2) by the first three dimensions. Mg was closely associated to Dim 2 (top left) while all remaining elements were clustered together and more closely associated with Dim 1.

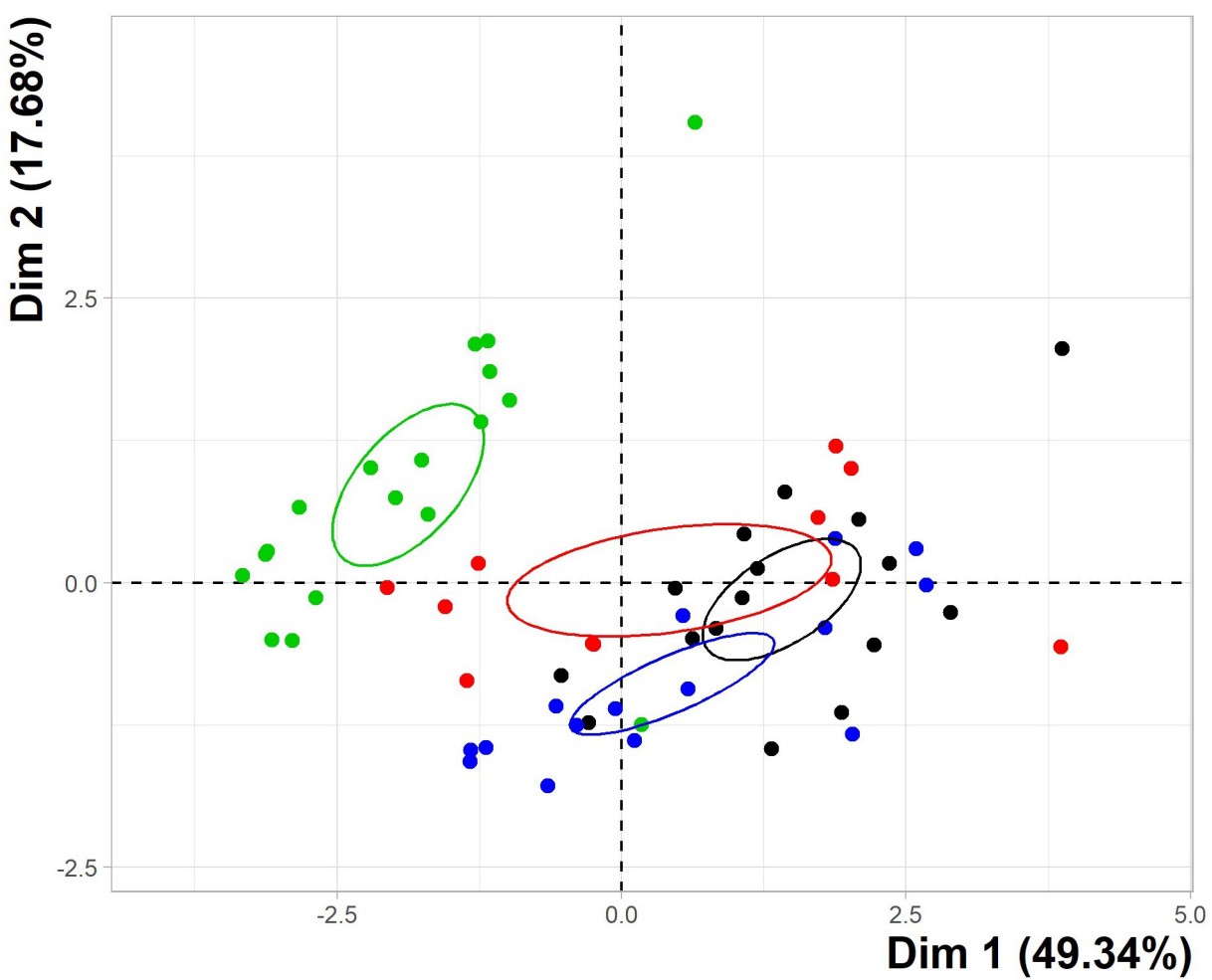

**Fig 3. PCA individual plot.** Depiction of all seagrass metal data reduced to two principal dimensions (Dim 1 and 2). Each point indicates an individual seagrass sample and data are categorised by location, represented as different colours (Howick Island Group, HWK = green, Cockle Bay, CB = black, Upstart Bay, UB = blue and Edgecumbe Bay, EB = red). Confidence ellipses are colour coordinated with the individual sample points. Dimension 1 (Dim 1) represents 49.34% of total variations and Dim 2 represents 17.68%. HWK data provided by Thomas et al [5].

**Table 2. Table of mean coastal seagrass metal concentrations before and after the wet season.** Mean concentration and standard deviation (mg/kg (dw) ± SD) of metal elements in seagrass samples collected at each coastal site (CB, UB and EB) before and after the 2018/19 wet season.

| | CB | | UB | | EB | |
|---|---|---|---|---|---|---|
| Element | Before | After | Before | After | Before | After |
| Al | 3290 ± 1250 | 4160 ± 1760 | 1730 ± 890 | 1470 ± 570 | 1850 ± 750 | 1950 ± 600 |
| Cd | 0.3 ± 0.1 | 0.5 ± 0.1 | 0.3 ± 0.1 | 0.4 ± 0.1 | 0.3 ± 0.1 | 0.3 ± 0.1 |
| Co | 1.7 ± 0.4 | 1.4 ± 0.5 | 1.9 ± 0.5 | 2.2 ± 0.6 | 1.4 ± 0.7 | 1.0 ± 0.4 |
| Cu | 5.6 ± 2.3 | 4.3 ± 0.8 | 5.0 ± 0.5 | 7.2 ± 2.3 | 3.0 ± 0.6 | 2.5 ± 0.7 |
| Fe | 3070 ± 920 | 3380 ± 1240 | 1950 ± 1150 | 3500 ± 1650 | 3120 ± 1090 | 2860 ± 1020 |
| Mn | 350 ± 73 | 240 ± 67 | 246 ± 110 | 300 ± 120 | 210 ± 110 | 120 ± 76 |
| Ni | 3.4 ± 0.7 | 3.9 ± 1.1 | 4.3 ±1.5 | 4.2 ± 1.3 | 3.9 ± 1.7 | 5.3 ± 1.5 |

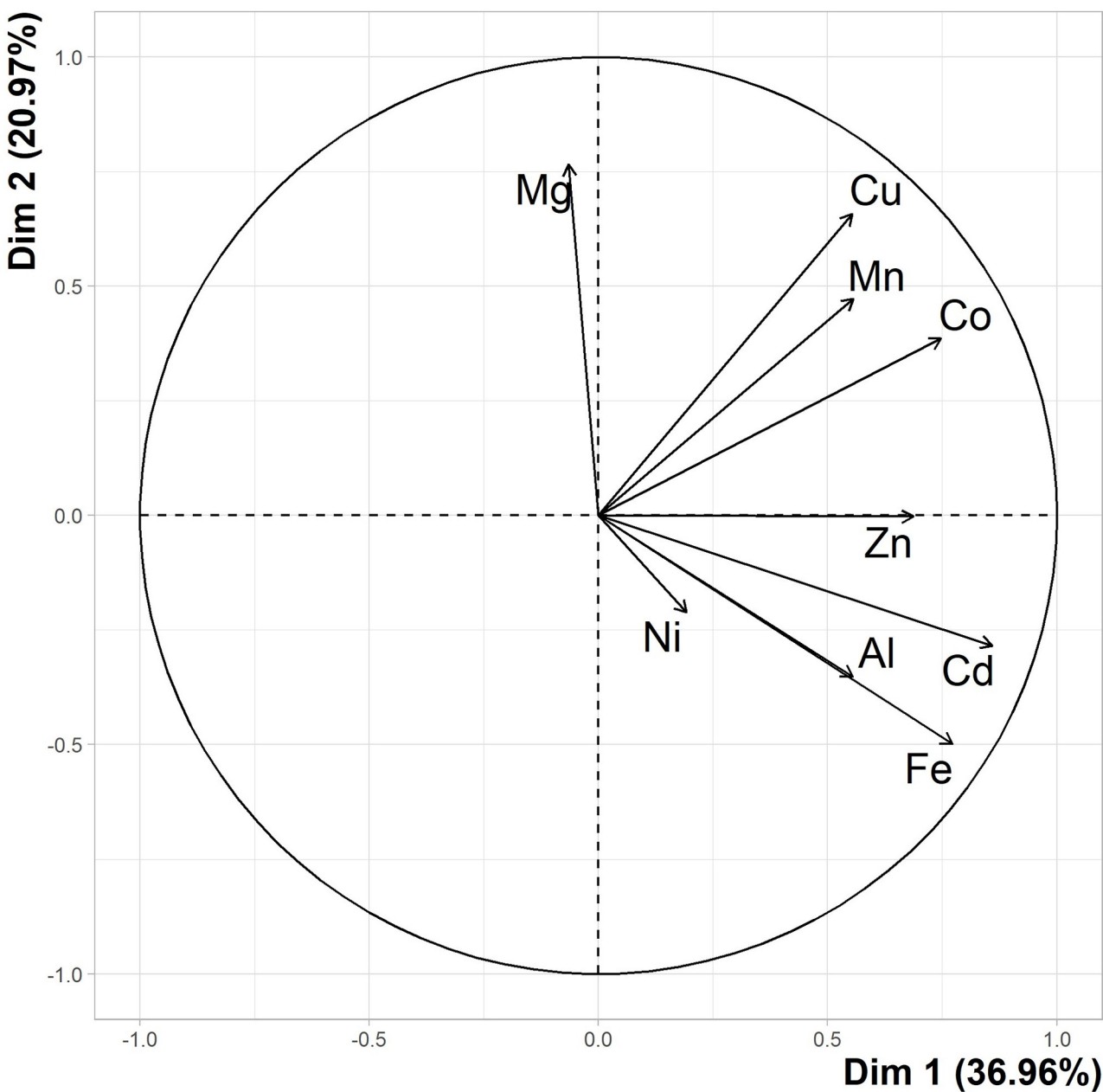

**Fig 4. PCA Variable loading plot of seagrass metals before and after the wet season.** PCA findings of all metal concentration data from seagrass collected from coastal sites of this study pre-wet and post-wet season. Dimension 1 (Dim 1) represents 36.96% of total variations and Dim 2 represents 20.97%.

Individual samples from both events were evenly distributed across the entire plot (Fig 5). The confidence ellipses demonstrate that data from both sampling events show some association with one another. While ellipses do not intercept or overlap, little space separates them and individual samples from both sets were located within both ellipses.

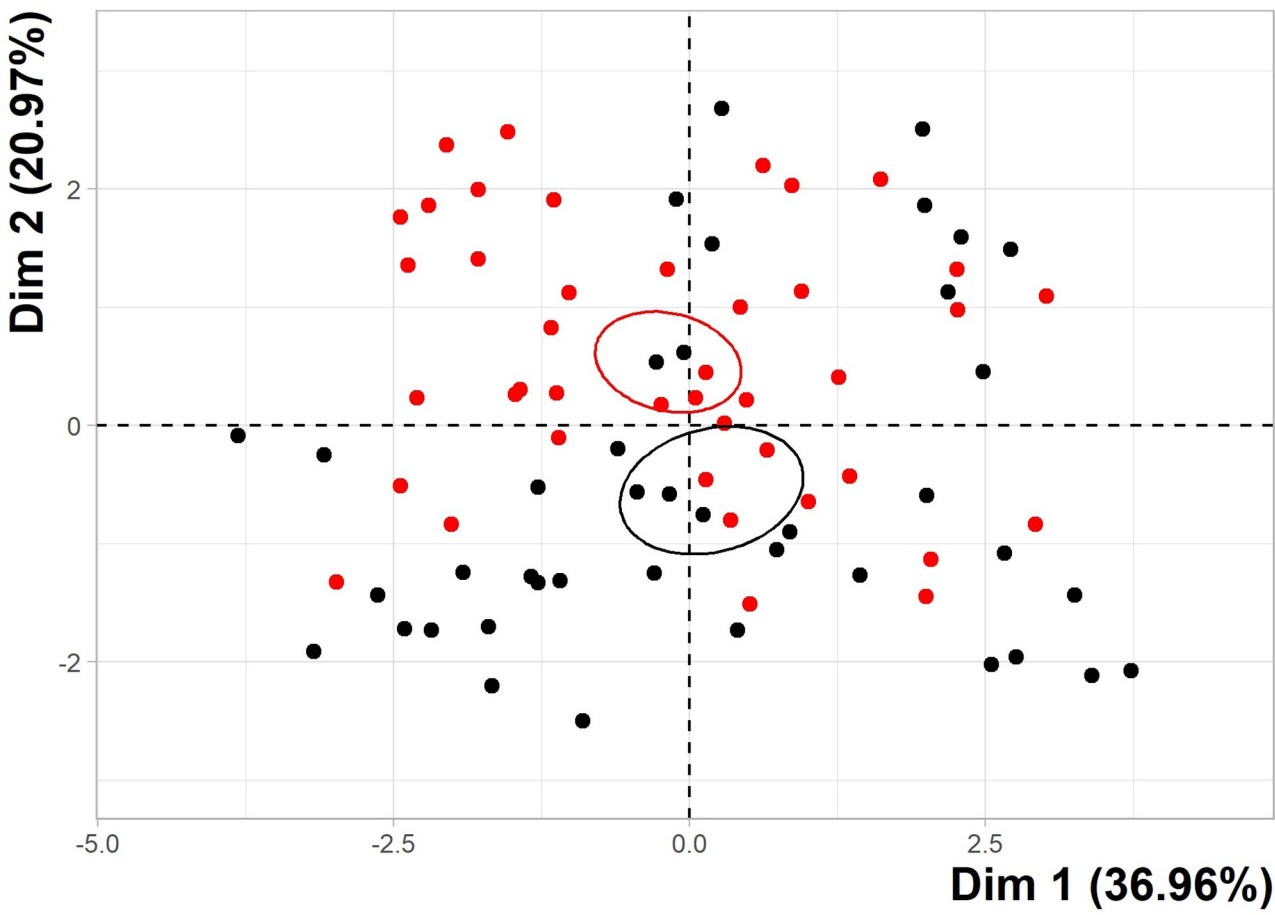

**Fig 5. PCA individual plot of metal concentrations before and after the wet season.** Depiction of all coastal metal data, collected in this study, reduced to two principal dimensions (Dim 1 and 2). Each point indicates an individual seagrass sample and are categorised by sampling event and categorised as different colours (pre-wet season = red and post-wet season = black. Confidence Ellipses are colour coordinated with the individual sample points. Dimension 1 (Dim 1) represents 36.96% of total variations and Dim 2 represents 20.97%.

## Discussion

### Total metal concentrations relative to calculated reference values

By analysing seagrass metals in coastal study sites alongside to an ecologically different natural baseline population that is minimally impacted by anthropogenic activity (in this instance HWK), some insight may be gleaned as to whether target elements were potentially detected at elevated levels within seagrass meadows in study sites close to human influence. Region-specific baseline data are crucial as reference for determining whether element concentrations are of concern to the local ecosystems or animals. Here Cd concentrations were greater at all three coastal sites relative to those reported in seagrass from HWK [5], possibly indicating exposure to higher concentrations at all coastal sites monitored. Additionally, coastal Cd levels exceeded, or were close to, offshore site seagrass data (0.36 mg/kg), reported by Conti *et al.* [43] in temperate seagrass species found in the Mediterranean Sea. One potential source of Cd in the region may be due to soil erosion of old Cd-rich coastal sugar cane paddocks, over time [44, 45]. Cd, in conjunction with many other metal elements (and other contaminants), can bind to suspended fine particulates in the Burdekin River, and be transported to the coastal zone

during outflow. This exposure pathway is particularly significant during major freshwater run-off events, and through increased erosion seen in the Burdekin catchment [46], which encompasses UB. Cd is considered a xenobiotic element (not produced or used within the organism), and it is possible that exposure to very low concentrations may cause some toxic effects [26]. Due to the low excretion rate and its ability to bioaccumulate in tissues, Cd is considered a high risk metal in terms of toxicity to organisms [47], including *C. mydas* [48].

Like Cd, seagrass Co concentrations were elevated in coastal sites, relative to HWK (Table 1), and most of all in UB (1.87 ± 0.45 mg/kg), approximately four times higher than HWK (0.52 ± 0.21 mg/kg). Co is deemed beneficial to plants as micronutrients and was reported to actively accumulate in seagrass leaves and roots [49, 50]. Co concentrations of up to four times higher than HWK have been reported in *C. mydas* within the region and are of some concern as immunosuppression has previously been linked to Co exposure in marine turtles [5, 51, 52]. Over recent years, consistently high concentrations have been reported in both *C. mydas* blood [51], and preferred forage samples [5] within the study region. Villa et al. [51] reported Co blood concentrations in UB *C. mydas* exceeded site-specific reference intervals (RIs) by up to an order of magnitude. These RIs were calculated to determine if any elements were considered elevated relative to a baseline cohort of clinically healthy turtles considered to be minimally impacted by anthropogenic chemical influence. The elevated Co concentrations are not believed to be due to any recent contamination event, rather concentrations are consistent over time. Upstart Bay receives river discharge from the Burdekin River, where historical land use in this region includes Ni and Co mining (Greenvale). Erosion rates in this region have significantly increased (by up to eight times) since the 1850s and is associated with rangeland beef grazing [45, 53]. Given that Co can be transported in particulate form associated with soil particles and marine sediments, it is plausible that changes in the supply of Co to the coastal region may be associated with increased erosion and transport of fine fractions of sediment [46]. Fine sediments (muds and silts) contain higher concentrations of certain metals relative to coarser types (sands and gravel) [54]. The increase in fine sediment discharge within the Burdekin may be associated with increased metal loads (particularly if concentrations were higher than previous discharge events), which bind to fine sediments. The majority (up to 67%) of sediment discharged from the Burdekin River has been found to deposit in UB with long-distance sediment transport also likely to CB somewhat [55]. This pattern is reflected in the current study, whereby forage Co concentrations are greatest at UB and decline (but are still high) in CB.

Comparisons between sites can be informative, though it is difficult to directly compare between sites due to differences in local conditions and metal geoavailability. Geochemistry and bioavailability of metals differs between region, zone and proximity to contaminant sources and are mediated by complex physical, chemical and biological interactions, which are largely poorly-understood [5]. Sediment type, texture and mineralogy all play a role in determining availability [5], with particles of fine clay and silt, found in estuarine environments (CB and UB), often carrying significantly greater terrigenous metal loads than coarser carbonate based sediments found offshore (where marine metals tend to be higher than in coastal samples), in areas such as HWK. Region-specific variations in sediment profile make it difficult, and often unreliable, to compare metal concentrations directly between sites, as definitive explanation for differences is often unknown. While such comparisons give insight into relative concentrations, they provide little information as to whether such concentrations are at normal loads for the region or if that site is in fact contaminated [5].

While the multivariate approach and PCA conducted was able to reduce the dimensions of the data from nine variables (metal elements) down to two, the results did not indicate any element or interaction of elements that influenced the differences in metal profiles between

coastal sites. All variables were loaded evenly and influenced the data set to similar extents. The PCA indicated that metal profiles were distinctly different between HWK and all coastal sites, with strong association between coastal profiles observed in the data. While coastal profiles (particularly CB) were like one another, differences were still found between locations, likely explained by region-specific geoavailability differences, whereby degree of anthropogenic influence and local environmental conditions and processes play important roles in the bioavailability and distribution of persistent chemical contaminants in local environments. Greatest mixing of individual sample data was between data from CB and UB, indicative of the most similarity. These sites are located close to one another and are influenced by the Burdekin River out flow, and thus share sediment sources [55, 56]. Furthermore, HWK data was most like EB (eclipse proximity). Out of all three coastal sites EB is likely the least influenced by anthropogenic impact as interpreted here (consecutively lower metal concentrations), when compared to CB and UB.

## Impacts of elevated or toxic metals on seagrass survivability

Metals such as Cu, Cd and Zn are thought to be toxic to seagrass species. In this study Cu concentrations were detected between 2.5 ± 0.7, in EB post-wet season and 7.6 ± 2.3 mg/kg, in UB post-wet season. Zheng *et al.* [57] observed leaf necrosis in *Thalassia hemprichii* after a 5-day treatment to 1 mg/L $Cu^{2+}$, likely a symptom of malnutrition, as competition for micronutrient uptake binding sites could induce inhibition of transport and function of ions such as $Ca^{2+}$ (calcium), $Mg^{2+}$, $K^{2+}$ (potassium), which are micronutrients required for numerous metabolic and photosynthetic processes [57, 58]. Furthermore, photosynthetic efficiency may be hindered by phytotoxic effects of some metals, including changes in redox states in leaf cells due to inhibition of antioxidant enzymes, such as superoxide dismutase and peroxidase, which leads to increased production of radical oxygenating species (ROS) that damage photosynthetic apparatus and chlorophyll [57]. Necrosis and antioxidant inhibition may be accompanied by a significant decline in photosynthetic efficiency (effective quantum yield) [57], which is likely caused by disturbance to photosynthetic electron transport observed in a range of contaminants including photosystem-II herbicides [59], also commonly applied in agricultural practices throughout the study region. Reduced photosynthetic function often leads to inhibited growth, survival, and community fitness of exposed meadows [60], leading to declines in distribution and inevitably habitat loss for the plethora of species reliant on seagrass ecosystems for a range of ecological functions.

## What elevated metal loads mean for *C. mydas*

Metal concentrations detected in *C.mydas* depend on numerous factors including, species, sex, age and location [61]. Element concentrations were detected at highest concentrations at coastal sites relative to HWK data, which implies the likelihood of increased exposure of local coastal foraging *C. mydas* to potentially toxic metals. Toxic metal exposure has been reported to impact different aspects of marine turtle physiology, immunology, and biochemistry [26, 51, 62–64]. One such impact is that of elements including Cd, Co, Cu, Fe and Ni and Pb which have previously been reported as potentially causing immunosuppression in individuals, leading to increased susceptibility to secondary infections which may also be associated as necessary factors in expression of the enzootic disease, Fibropapillomatosis (FP), in *C.mydas* [26]. Though due to fundamental ethical issues, toxicity threshold data has not been calculated for marine turtles for any metals and thus limited information is available on what concentrations of metals (and other contaminants) are of ecological significance to local *C. mydas* health. One recent advancement and one which requires further application, is that of cell-based toxicity

assays which have been applied to measure ecotoxicological endpoints of suites of metal elements on cultured *C. mydas* cells [65–67]. Continued effort should be made to implement such *in vitro* approaches alongside conventional chemical analysis to better understand metal specific toxicities and to calculate site-specific toxic thresholds which may be implemented to better determine the exposure and susceptibility of local foraging populations.

## Variation in metal concentrations following the wet season

In UB for most trace elements, increases in concentrations, relative to HWK data, were reported in samples collected pre-wet season, whereas in CB, the opposite was true. This is interesting as during January and February 2019 significant rainfall (a total of 1260 mm over ten days) caused an extreme flood event in the Townsville region, which exceeded historical records (926 mm over ten days in 1953) [68]. This event impacted numerous areas along the coast, adjacent to the study region (except HWK). Large flood plumes entered the coastal zone from the Ross River (Townsville) and the Burdekin River. Increased terrigenous sediment and metal transport into the coastal marine environment during major flood events has previously been reported [69], whereby fluvial transport processes cause sedimentary grain sizes to be sorted by size with increased distance from a given source [69]. The high affinity of trace metals for fine sediment particles means that the transport of metal loads often follows flood plume patterns, settling in low energy environments such as sheltered bays [70, 71], like CB and UB. However, suspended metal loads tend to be taken up by phytoplankton plankton blooms prior to settlement [72] and thus concentrations which are available for uptake by seagrass may be reduced. Furthermore, the 2019 major flood event may have brought increased metal loads to the study sites but were still sequestered in sediments and not yet accessible to seagrass.

## Conclusion

Across the 2018/19 wet season, trace metal concentrations were measured in preferred (*C. mydas*) seagrass forage species. Several elements were found at greater concentrations in seagrass from the Northern and Central GBR region, when compared to site-specific reference data, in both coastal and offshore sites with various levels of anthropogenic influence. Elements of most concern, Cd and Co (thought to be toxic to seagrass and *C. mydas* alike), were found at greater concentrations in samples collected from all coastal sites when compared to the offshore site. Additionally, the Cd and Co offshore concentrations both exceeded published reference data, though definitive conclusions could not be drawn regarding the threat posed by these concentrations, to marine turtle health as site-specific reference data are lacking for these metals. Additionally, no distinct patterns were observed in metal concentrations detected in seagrass samples, from any sites, collected prior to the 2018/19 wet season when compared to data collected post-wet season. This was likely due to processes not within the scope of this descriptive study and thus not investigated. Metal concentration comparisons between study sites offer some information on local metal loads. For instance, PCA determined that coastal metal profiles were more like one another than to the offshore site but should be analysed tentatively as environmental factors such as sediment type geomorphology and geochemistry play a role in determining the geoavailability of certain elements. A better approach is to compare data to site-specific baseline values from that same location to provide insight into whether current levels are of concern. A significant increase in funding and investigation into the ecotoxicological study of environmentally relevant metals and the potential sources of such chemical contaminants is crucial before any insight can be gleaned regarding what the current

metal loads likely mean for local seagrass meadows and the macrograzers which rely on them as forage.

## Supporting information

**S1 Fig. RAW data analysed and reported in research article "Trace element concentrations in forage seagrass species of *Chelonia mydas* along the Great Barrier Reef".**
(DOCX)

## Author Contributions

**Conceptualization:** Adam Wilkinson, Ellen Ariel, Jon Brodie.

**Data curation:** Adam Wilkinson.

**Formal analysis:** Adam Wilkinson.

**Funding acquisition:** Adam Wilkinson, Ellen Ariel, Jon Brodie.

**Investigation:** Adam Wilkinson, Ellen Ariel.

**Methodology:** Adam Wilkinson, Ellen Ariel, Jason van de Merwe, Jon Brodie.

**Project administration:** Adam Wilkinson.

**Resources:** Adam Wilkinson, Ellen Ariel.

**Software:** Adam Wilkinson, Jason van de Merwe.

**Supervision:** Ellen Ariel, Jason van de Merwe, Jon Brodie.

**Visualization:** Adam Wilkinson.

**Writing – original draft:** Adam Wilkinson.

**Writing – review & editing:** Adam Wilkinson, Ellen Ariel, Jason van de Merwe, Jon Brodie.

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
