## [Decision Letter · Decision Letter 0]

2 Feb 2022

PONE-D-21-24131Trace element concentrations in forage seagrass species of Chelonia mydas along the Great Barrier ReefPLOS ONE

Dear Dr. Wilkinson,

Thank you for submitting your manuscript to PLOS ONE. After careful consideration, we feel that it has merit but does not fully meet PLOS ONE’s publication criteria as it currently stands. Therefore, we invite you to submit a revised version of the manuscript that addresses the points raised during the review process.

We look forward to receiving your revised manuscript.

Kind regards,

Mohan Lal Dotaniya, Ph.D.

Academic Editor

PLOS ONE

Journal Requirements:

3. We note that Figure 1 in your submission contain map images which may be copyrighted. All PLOS content is published under the Creative Commons Attribution License (CC BY 4.0), which means that the manuscript, images, and Supporting Information files will be freely available online, and any third party is permitted to access, download, copy, distribute, and use these materials in any way, even commercially, with proper attribution. For these reasons, we cannot publish previously copyrighted maps or satellite images created using proprietary data, such as Google software (Google Maps, Street View, and Earth). For more information, see our copyright guidelines: http://journals.plos.org/plosone/s/licenses-and-copyright.

Additional Editor Comments (if provided):

Manuscript falls under aim and scope of the journal.

Please follow the journal guidelines and address the reviewers comments properly.

Reviewers' comments:

Reviewer's Responses to Questions

**Comments to the Author**

1. Is the manuscript technically sound, and do the data support the conclusions?

Reviewer #1: Yes

Reviewer #2: Yes

2. Has the statistical analysis been performed appropriately and rigorously? 

Reviewer #1: Yes

Reviewer #2: Yes

3. Have the authors made all data underlying the findings in their manuscript fully available?

Reviewer #1: Yes

Reviewer #2: Yes

4. Is the manuscript presented in an intelligible fashion and written in standard English?

Reviewer #1: Yes

Reviewer #2: Yes

5. Review Comments to the Author

Reviewer #1: This is a solid study requiring only minor editorial revisions.

Reviewer #2: Dear Sir,

Greetings!

The manuscript entitled “Trace element concentrations in forage seagrass species of Chelonia mydas along the Great Barrier Reef” is written well and has the scientific importance. There are certain suggestions and question from my side.

## Abstract is written well but it needs to make it concise for a gist. The sentence Toxic metal exposure is a threat to green sea turtles (Chelonia mydas) inhabiting and foraging in coastal seagrass meadows and is of particular concern in local bays of the Great Barrier Reef (GBR), as numerous sources of metal contaminants are located within the region. Seems to Introductory sentence.

## Due to this, the primary source of metal element exposure for green turtles (Chelonia mydas) is through their diet. make it clear

##In addition to providing forage for C. mydas, seagrass fulfil several integral ecological 61 functions, such as sediment stabilization nutrient cycling the sequestration of 62 carbon Put the latest and relevant reference.

##Gastric lavage sampling on a subsection of each population elaborate it?

##Once identified, seagrass samples were analysed for a suite of 10 metal elements using Inductively coupled Plasma Optical Emission Spectrometry to describe concentrations detected within coastal meadows along the Central GBR. Put the Modal and reference of analysis protocol.

## Fig 1 images are having poor visualization need to be update. Other figures are also poor in quality.

## The finding should be supported with latest reference there is no reference from 2021 and 2022

##Conclusion Section needs to be summaries well.

##Please double-check the manuscript for abbreviations. Abbreviations must be spelled out the first time they are mentioned in the abstract and starting again with the introduction section. Double-check that all references are cited within the text, and that all citations within the text have a corresponding reference. Double-check the spelling of the author names. Please ensure the abbreviation of the journal name.

6. PLOS authors have the option to publish the peer review history of their article (what does this mean?). If published, this will include your full peer review and any attached files.

Reviewer #1: No

Reviewer #2: **Yes: **Hanuman Singh Jatav

---

## [Author Response · Author response to Decision Letter 0]

15 Apr 2022

Responding to: Point raised Action taken? Response

Academic editor Manuscript formatting check YES I checked all formatting of the manuscript. I did not find guidelines regarding file naming in either of the formatting PDFs or the PLOS one submission guidelines.

Academic editor Ethics statement must be in Methods section YES I moved the ethics statement to the methods section 

Academic editor Fig 1 map image citation YES Actions outlined in Esri ArcGIS permissions for use were adhered too. Acknowledgement of use of Arc GIS and base map citation included in figure caption. Specific written permission is not necessary to use a map created in ArcGIS software.

Academic editor Captions for supporting information YES I added the title and caption to the supporting information, added the supporting information section at the end of the manuscript and cited the supporting information in the text as requested. 

Academic editor Review references YES All references were reviewed and are all cited within the text. No removal of references was conducted.

Reviewer #2 Abstract is written well but it needs to make it concise for a gist. The sentence Toxic metal exposure is a threat to green sea turtles (Chelonia mydas) inhabiting and foraging in coastal seagrass meadows and is of particular concern in local bays of the Great Barrier Reef (GBR), as numerous sources of metal contaminants are located within the region. Seems to Introductory sentence. YES The highlighted sentence was moved and the abstract was altered to provide more of a concise read.

Reviewer #2 Due to this, the primary source of metal element exposure for green turtles (Chelonia mydas) is through their diet. make it clear YES The statement in question was altered for clarification

Reviewer #2 In addition to providing forage for C. mydas, seagrass fulfil several integral ecological 61 functions, such as sediment stabilization nutrient cycling the sequestration of 62 carbon Put the latest and relevant reference. YES 2021, 2022 or other newer relevant references were added where available and required.

Reviewer #2 Gastric lavage sampling on a subsection of each population elaborate it? YES The statement was altered to provide clarification

Reviewer #2 Once identified, seagrass samples were analysed for a suite of 10 metal elements using Inductively coupled Plasma Optical Emission Spectrometry to describe concentrations detected within coastal meadows along the Central GBR. Put the Modal and reference of analysis protocol NO No amendment was made in the section of note as the model and analysis protocol was later referenced in a more appropriate methods section.

Reviewer #2 Fig 1 images are having poor visualization need to be update. Other figures are also poor in quality. NO The image quality of all figures is within the 300-600 dpi guideline for the journal and no comment on this was made by the academic editor or reviewer #1

Reviewer #2 The finding should be supported with latest reference there is no reference from 2021 and 2022 YES 2021, 2022 or other newer relevant references were added where available and required

Reviewer #2 Conclusion Section needs to be summaries well. YES The conclusion has been added to and re-written to better convey the findings of the study and to summarise it best.

Reviewer #2 Please double-check the manuscript for abbreviations. Abbreviations must be spelled out the first time they are mentioned in the abstract and starting again with the introduction section. YES The manuscript was checked for all abbreviations. All uses of abbreviations were correctly used following the first instance of being mentioned.

Reviewer #2 Double-check that all references are cited within the text, and that all citations within the text have a corresponding reference. Double-check the spelling of the author names. Please ensure the abbreviation of the journal name. YES All references and citations were checked against the reference list and all are correct. Journal names were abbreviated for those references not already done so

---

## [Editor Report · Decision Letter 1]

31 May 2022

Trace element concentrations in forage seagrass species of Chelonia mydas along the Great Barrier Reef

PONE-D-21-24131R1

Dear Dr. Wilkinson,

We’re pleased to inform you that your manuscript has been judged scientifically suitable for publication and will be formally accepted for publication once it meets all outstanding technical requirements.

Kind regards,

Mohan Lal Dotaniya, Ph.D.

Academic Editor

PLOS ONE
---

## [Editor Report · Acceptance letter]

6 Jun 2022

PONE-D-21-24131R1 

Trace element concentrations in forage seagrass species of *Chelonia mydas* along the Great Barrier Reef 

Dear Dr. Wilkinson:

I'm pleased to inform you that your manuscript has been deemed suitable for publication in PLOS ONE. Congratulations! Your manuscript is now with our production department. 

Kind regards, 

on behalf of

Dr. Mohan Lal Dotaniya 

Academic Editor

PLOS ONE